# The Impact of Working Memory on the Development of Social Play in Japanese Preschool Children: Emotion Knowledge as a Mediator

**DOI:** 10.3390/children10030524

**Published:** 2023-03-08

**Authors:** Hisayo Shimizu

**Affiliations:** Graduate School of Humanities and Social Sciences, Hiroshima University, Higashihiroshima 7398524, Japan; hisayos@hiroshima-u.ac.jp

**Keywords:** working memory, emotion knowledge, social play, preschoolers, pediatric neurology, pediatric psychology, social-emotional competence, early childhood development, Japan, Play Observation Scale

## Abstract

Through enriched play, children learn social-emotional skills necessary for academic achievement and interpersonal relationships with others. Further research is needed on how specific factors associated with social play, such as working memory and emotion knowledge, interact to promote it. Previous studies have examined the association of working memory and emotion knowledge with social play. However, there are no consistent results as to which abilities influence which skills first. Thus, the present study examines the impact of working memory on the development of social play and the role of emotion knowledge in the relationship between working memory and social play. Forty-seven Japanese preschoolers were tested on working memory, social play, and emotion knowledge. Regression analysis indicated that working memory was significantly related to social play. Furthermore, mediation analysis indicated that emotion recognition mediates the effects of working memory on social play. Working memory was found to contribute to social play by improving emotion recognition in children. These results indicate that the pathway from working memory to social play is mediated by emotion recognition and expands previous perspectives on the developmental mechanisms of emotion knowledge in children.

## 1. Introduction

Play is essential for children’s development of social-emotional competence in early childhood. In particular, social play has been linked to the development of executive function [1], emotion knowledge [2,3], theory of mind [4], social competence [5], and psychosocial adjustment [6], including anxiety and problem behaviors [7], reading comprehension, and math performance [8]. Successful peer relationships in early childhood are also related to academic achievement and school adjustment beyond elementary school [9,10,11,12,13]. Researchers have found that, among other things, the development of executive function and emotion knowledge has an important influence on socioemotional competence in early childhood [14,15]. However, it is unclear whether one competence developmentally precedes the other. Since play is considered the foundation of the early childhood curriculum [16], it is necessary to examine how executive function and emotion knowledge are related to it.

Social play includes a child’s ability to interact with their peers, including behaviors such as collaboration, imaginative play, and role-shifting play, which surpasses the activity of just playing with others [17]. Children who interact more with their peers have been reported to have higher language and math skills, less aggressive behavior [18], and better relationships with their teachers [8]. Children with higher levels of imaginative play have been shown to have higher executive function, emotion knowledge, language skills, and more conversations with peers [2,14]. In addition, pretend play is associated with inhibitory function [1,19,20] and social skills [5]. Play with dolls, compared to play with a tablet, may provide more opportunities to rehearse theory of mind and empathy skills and contribute to the development of social-emotional skills [21,22,23]. Social play in early childhood is thought to promote the acquisition of others’ perspectives, conflict resolution, and interpersonal problem-solving skills [23,24,25]. Thus, social play significantly predicts higher social skills, reduced rates of social isolation, and improved social competence in interpersonal relationships, including with peers and adults [26,27,28]. Therefore, investigating the predictors of social play is important for helping adults understand how to help their children develop social abilities that will contribute to their psychosocial adaptation.

Executive function (EF) is an ability associated with children’s social play [29]. EF is a self-regulating cognitive process that controls thoughts, behaviors, and emotions to achieve specific goals. EF develops rapidly in early childhood. Its development not only affects early childhood, but also continues into adolescence. Furthermore, the EF is strongly associated with the prefrontal cortex (PFC): EF development is maintained by the growth of neural networks that include the PFC [30]. Children with high executive function have been shown to have better emotional control and less aggressive behavior [31]. Executive function and self-regulation skills have also been shown to interact and have indirect effects on academic achievement and social skills [32,33]. Furthermore, a link between executive function and theory of mind has also been noted [34]. Early executive function predicts theory of mind, and as children age, executive function and theory of mind have been shown to have reciprocal effects [35]. In addition, children with low executive function have been noted to have problems with school adjustment [36], including hyperactivity, inattention, and impulsivity, as well as lower academic achievement [37]. Children’s EF has an essential role in prosocial behavior, better concentration, self-regulation skills [32], theory of mind [34], academic performance, and school readiness [37]. EF comprises working memory, cognitive flexibility, and inhibitory control [38].

Working memory is a basic EF and is defined as the ability of the mind to retain and manipulate information for complex cognitive tasks [39]. Children rated as having low working memory have been found to relate to poorer academic performance [40,41] and lower intelligence [42] compared to those of the same age. Working memory also facilitates children’s development of their own understanding of their mental states, including false beliefs [43] and emotional understanding [44]. When children engage in play, their working memory processes allow them to understand their peers’ perspectives or maintain consistency in play rules and scenarios. Thibodeau et al. showed that children whose working memory was improved by imaginative play became more engaged in pretend play [45]. Thus, working memory facilitates children’s social play by focusing on achieving social goals, retrieving information from retained memories, and selecting behavioral responses based on appropriate information. In addition, some studies suggest that working memory has a critical role in children’s emotion knowledge [43,46]. The main reason working memory is essential for emotion knowledge is that the cognitive scheme for understanding emotion-provoking situations is a complex structure that links emotions, other mental states (such as goals, desires, and expectations), and external events and contextual representations (such as goal achievement and failure situations) [47]. There is evidence that working memory capacity affects the development of emotion knowledge [39] and that working memory predicts emotion knowledge [48].

Children’s emotion knowledge is their ability to identify emotions from facial expressions, understand situations that trigger emotions, and acquire emotional language [49]. For example, a child who successfully engages in peer play will be able to notice and correctly understand their peers’ emotional expressions. Such children are then able to identify their peers’ emotional expressions in the context of play and regulate their own expressions accordingly. In terms of the normative development of emotional knowledge, between the ages of 1.5 and two years, children can label their emotions with words, albeit less frequently [50]. Children under two years will generally be able to identify whether they are feeling good or bad [51]. The ability to recognize and label basic emotions (happy, sad, angry, scared) emerges between the ages of two and three, and the ability to recognize basic facial expressions becomes proficient by the age of five years [52]. The ability to understand the context surrounding a particular emotion, i.e., being happy when receiving a cookie or sad when losing a toy, is developed by the end of kindergarten [53,54]. Most children can acquire the ability to label emotions and recognize situations that elicit emotions by the age of five years. However, there are individual differences regarding the development of these abilities [55].

Emotion knowledge has two aspects: the understanding of emotional expression (the ability to discern emotions from facial expressions) and the recognition of emotion-provoking situations (the ability to identify the original emotions elicited by specific situations) [56]. After infants have developed a basic level of emotion recognition ability, they acquire emotion situation knowledge and are able to understand the situation or context in which the emotion occurs [10,57]. Regarding the relationship between social play and emotion knowledge, observations have revealed that children with higher levels of emotion situation knowledge spend more time on social play [58]. Social play persists because children who understand the social context and can correctly interpret emotional cues can determine whether their entry into play will be successful or rejected [59].

While emotion knowledge predicts higher social competence, higher quality peer relationships, higher academic achievement, and fewer behavioral problems, some research suggests that it is EF that predicts the early development of emotion knowledge [60,61]. Some studies have also shown that low levels of preschool EF predict difficulties with behavioral adjustment in school, especially with hyperactivity, even if the child has moderate social-emotional competence [62]. Thus, some studies suggest that EF is a predictor of emotion knowledge, while others suggest that EF and emotion knowledge interact with each other [15]. Farrell and Gilpin’s study showed that emotion knowledge and EFs develop interactively from preschool through kindergarten [15], which means that the ability to identify emotions may help one to inhibit one’s emotions when encountering emotionally challenging situations, and the opposite is true as well.

Though the association between social play, working memory, and emotional understanding has been clarified, efforts to determine the mechanisms of the association between social play, working memory, and emotional understanding are limited. This study aimed to explore the mediating role of emotion knowledge in the association between young children’s working memory and social play. The following three hypotheses were developed.

Working memory is positively correlated with emotion recognition and emotion situation knowledge.Emotion situation knowledge is more sophisticated than emotion recognition and, thus, is more strongly associated with social play.Emotion knowledge mediates the relationship between working memory and social play.

This study uses nature observation to systematically observe children’s social play. Teacher-reported preschooler social play is a more global aspect of children’s social engagement behavior and may be influenced by children’s non-free play adaptive behaviors, academic performance, and teachers’ perceptions of the parent-child relationship. Teachers find keeping track of children’s social play difficult because children often engage in it without a teacher nearby. As such, using nature observation to systematically observe children’s social play can help avoid problems with teacher evaluation.

Japanese young children tend to show fewer negative emotions in interpersonal conflict situations than American children [63,64]. These differences are consistent with the Japanese taboo of expressing negative emotions in public [65]. Given this cultural background, Japanese preschoolers’ ability to understand emotions is considered essential for successful social play in free-play situations.

## 2. Materials and Methods

### 2.1. Participants

Forty-seven Japanese children participated in this study, including 17 four years old (*M* = 55 months, *SD* = 2.33), 26 five years old (*M* = 64 months, *SD* = 2.84), and four six years old (*M* = 72 months, *SD* = 0.00). The sample comprised 53% boys. Participants were native Japanese speakers recruited from two classes of nursery schools located in the Kinki region of Japan. Detailed demographic information on the children was not available due to the school’s privacy policy. However, they were all from middle-class families. 

### 2.2. Procedure

Children were observed in a natural setting for three months and individually assessed by their teachers. The data collected included observations of children’s social play and direct assessments of their verbal and emotion knowledge abilities, as well as their working memory. At the beginning of data collection, participating schools sent home a letter explaining the study and requesting parental consent. Children with parental consent were asked to give their verbal consent before undergoing direct assessment.

Children’s play behavior in natural settings was observed three times a week. Assessments of individual children were conducted in a quiet corner of the child’s classroom, usually for 15 min, with tasks related to language ability, emotion knowledge, and working memory. Several graduate student research assistants who had undergone two weeks of training conducted naturalistic observations of children’s play administered the Picture Vocabulary Test-Revised and performed emotion knowledge and working memory assessments.

### 2.3. Measures

#### 2.3.1. Social Play

Trained research assistants observed the children both in the classroom and on the playground during the classes’ free-play times. They observed the children individually, in random order, during free-play for a total of 20 min (10 min per day, one session per week), with a total of 120 coding intervals (60 coding intervals per session). This procedure was used in a previous study, which found that observed nonsocial play was related to children’s psychological characteristics and the quality of mothers’ parenting behavior [66].

The observers coded social play with reference to the Play Observation Scale [67]. Interobserver reliability data were acquired from two observers, using 20% of the observations obtained. Observations of social play (K = 0.88) were used in the analysis.

Social play was coded using a time-sampling method when the target child was interacting with other children. Examples of behaviors that were coded as social play were pretend play, block play with two or more children, interactive play, and turn-taking. Social play, which did not include aggressive behaviors such as tapping and yelling, was coded when the children were engaging in interactions appropriately. Isolated behavior was coded when the targeted child was playing alone with play equipment or toys, observing other active children but not participating in their activities or walking around aimlessly. Relative frequency values for social play were calculated for each child through a ratio score by dividing the total number of observations of social play by the total number of observations overall.

#### 2.3.2. Emotion Knowledge

The children’s emotional knowledge was assessed with the Affect Knowledge Test [68,69], a measure that is commonly used to evaluate preschoolers’ emotional understanding. This assessment measures verbal expressive recognition, nonverbal receptive recognition, and situational knowledge. In the expressive recognition task, each child was presented with four emotion faces (i.e., happy, sad, angry, and afraid). Pointing to the face expressing each emotion, the researcher asked the child, “What is this face feeling?” In the receptive recognition task, the researcher shuffled the faces that represented the emotions and asked the child, “Where is the sad face?” Thus, the task involved both verbal expressive and nonverbal comprehension skills. Next, in the situational judgment task, four emotion faces were displayed to the child. The experimenters used puppets to read some stories about things that they might encounter in their daily lives to the preschoolers. Each child was asked how the character felt: “How does he or she feel?” The researcher asked the child to choose from among the four emotion faces to answer; thus, a verbal response was not required. The stereotyping task investigated the extent to which children could determine the feelings of others by using situations in which many people have similar feelings, such as feeling scared when they have a nightmare [69,70]. The non-stereotypical scenarios task measured how well the children were able to identify situations in which others had different feelings than the target child. Every situation elicited one of two emotions: feeling happy or sad about going to kindergarten. We also had conversations with the children beforehand and used the information we obtained to assess their ability to see others’ perspectives (e.g., showing that the main character is sad in a situation where the child is usually angry). Children’s performance was rated as 0 for incorrect responses and 1 for correct responses to all items. In this study, the emotion recognition score was an aggregate of the expressive and receptive recognition item scores. The situation knowledge score was a combination of the stereotypical and non-stereotypical situational knowledge item scores.

#### 2.3.3. Working Memory

The Backward Word Span [71] was utilized to assess children’s working memory. In this task, children were required to present a list of semantically unrelated words in monosyllables, repeating them in reverse order. Here, a lion puppet was used to demonstrate how to say the words in the backward order, e.g., the experimenter said, “pen, table” and the lion puppet replied, “table, pen”. This activity was followed by exercises (which were modified as needed) and test questions. The size of the list was increased with each successful trial (two, three, and four words). The score was the maximum number of words the child could reproduce, ranging from 1 to 5 points.

#### 2.3.4. Verbal Ability

The Picture Vocabulary Test-Revised [72] was utilized to assess children’s verbal ability. Four colored drawings were printed on a page, and the children were asked to select the one that best represented the stimulus word. In our analysis, we used the standardized scores of the Picture Vocabulary Test-Revised.

Child gender, age, and verbal ability were included in the model. Gender of the child was dummy coded (1 = male, 0 = female).

### 2.4. Data Analysis

The analyses were calculated with SPSS ver. 28. The descriptive statistics, means, standard deviations, and Pearson correlations were calculated for the demographic and study variables. To assess if emotion knowledge mediates the relationship between social play and working memory, the Hayes PROCESS method [73] (Model 4) was used, with 95% confidence intervals and a 5000-bootstrap sample. This procedure calculates the mediation effect according to the theoretical foundation established by Baron and Kenny [74]. 

## 3. Results

### 3.1. Descriptive Statistics

The means, standard deviations, and ranges of scores for age, gender, verbal ability, working memory, emotion knowledge, and social play are shown in Table 1. The correlations between these variables are shown in Table 2. Age was correlated with verbal ability and social play. The verbal ability was correlated with working memory, emotion recognition, and social play. Working memory was correlated with emotion recognition and social play. Emotion recognition and social play were strongly correlated, and situation knowledge was correlated with social play.

### 3.2. Mediational Analysis

As presented in Table 2, there were no significant correlations between working memory and situation knowledge. Therefore, a mediation model was developed using the bootstrap method to test the significance of the indirect effect of emotion recognition (mediator) on social play (dependent variable). In this model, gender, age, and verbal ability were used as control variables considering the established relationships between emotion knowledge and social play [58,75].

#### Social Play and Working Memory

As shown in Figure 1, in a mediation model of working memory and emotion knowledge as a predictor of social play, working memory had a significant indirect effect on social play through emotion recognition. Using the bootstrap method to test the significance of the indirect effect, the indirect effect was 0.06 (*p* < 0.001, 95% CI [0.002, 0.13]). In particular, the effect of working memory on social play when emotion recognition was controlled for was smaller than when it was not (direct effect of X: b = 0.07, *p* = 0.13; total effect of X on Y: b = 0.13, *p* < 0.01). In summary, higher levels of working memory predicted higher levels of social play and emotion recognition. Emotion recognition mediated the relationship between working memory and social play: The development of emotion recognition was related to improved social play.

## 4. Discussion

This study focused on the role of working memory in social play and examined whether emotion knowledge mediates the association between working memory and social play in children. The study found evidence to support previous findings regarding the association between working memory, emotion knowledge, and social play. First, regression analysis showed that working memory was positively correlated with social play. In addition, mediation analysis showed that emotion recognition mediates the relationship between working memory and social play. In other words, early working memory contributes to social play by improving children’s emotional awareness. The findings expand previous perspectives on how children’s social play develops and demonstrate that emotion knowledge is an important influence in its development. 

In the present study, the association between emotion recognition and working memory and the mediating effect of emotion recognition in the association between working memory and social play was stronger than that of situation knowledge. This result differed from those of previous studies that have emphasized the importance of situation knowledge [58]. This discrepancy may be due to cultural differences in understanding emotions between Japanese and Western populations. American culture is an independent culture where individual autonomy is valued. Thus, children are expected to express their feelings openly and directly [76]. In contrast, Japanese culture is an interdependent culture that emphasizes group harmony, leading people to suppress emotional expression, especially the expression of negative emotions [77,78]. Japanese mothers neither explicitly instruct nor encourage their children to express their emotions through language; they let the children learn naturally. The same approach is used in early childhood education in Japan. Japanese caregivers rarely work with children verbally to promote the development of emotional understanding of the self and others. Therefore, children learn implicitly about emotional understanding through interpersonal relationships. American parents and caregivers often recommend that children express the emotions they experience in words. However, this practice is uncommon in Japanese culture. 

Because Japanese preschoolers must learn about their own and others’ emotional states independently, individual differences in emotion recognition development may occur due to factors, such as the availability of learning opportunities. In fact, the emotion recognition scores were lower and more varied than the situation knowledge scores in this study. Other studies have also indicated that recognizing emotions takes time for Japanese preschoolers [63]. For example, a child who wants to play successfully with peers must use emotional knowledge to recognize the emotions of the other children in the group. Then, within the context of play, they learn to understand the emotional expressions of their peers and learn under what circumstances certain emotions are likely to arise. Emotion recognition is a foundational level component of emotion knowledge. However, among the Japanese preschoolers in this study, children with higher emotion recognition were more active participants in social play, suggesting that more frequent participation in play further promotes working memory and emotion knowledge.

Social play can explain the relationship between working memory and emotion knowledge [79,80]. As children’s social play activities require them to act toward a goal, social play requires working memory and emotion knowledge, which are trained during this play. Thus, social play helps children adapt to a constantly changing environment and not only improves working memory, but also helps in the development of emotion knowledge. This conclusion is consistent with those of other studies that have shown that higher working memory capacity is related to higher performance in emotion knowledge [39,81,82].

The present study shows that working memory influences social play by improving emotional understanding in children. Although previous studies have suggested a link between working memory, social play, and emotion knowledge, the mechanisms linking them have not been clarified. This study advanced the findings in this area by demonstrating that emotional knowledge plays a mediating role in the relationship with social play. To promote the development of preschoolers’ emotion knowledge, children must engage in planned play and activities that improve and develop their working memory. Children’s emotion knowledge develops through working memory, and in the process, their ability to understand emotions improves. The development of emotion recognition is especially important for Japanese preschool children, who need opportunities to name and remember emotional states experienced by themselves and others. This indicates the importance of Japanese parents and caregivers becoming actively involved linguistically to promote their children’s emotion recognition.

This study’s findings showed that working memory directly predicts social play. They also revealed that working memory indirectly supports social behavior by improving emotion knowledge, expanding on previous perspectives on how children’s social play develops. Nevertheless, the study has several limitations. First, asserting a causal relationship between working memory and social play was difficult because of the cross-sectional study design. Therefore, longitudinal studies are necessary to verify the causal relationship between working memory and social play that is inferred by the results of the present study. Second, the age of the children who participated should be considered. The sample in this study consisted of comparatively older children (mainly four and five years old). This factor may have resulted in a ceiling effect for emotion situation knowledge scores, leading to a lack of association with situation knowledge. In children younger than our sample (two or three years old), situation knowledge may be a stronger predictor of social competence. Finally, the sample size may have been insufficient. However, the moderate to large associations this study found between emotional knowledge and social behavior, as well as the mediation effect, reinforce the validity of its findings. Future studies should test the relationships with more evenly distributed and larger sample sizes.

## 5. Conclusions

This study demonstrates that working memory influences children’s social play and that emotion recognition plays a mediating effect. The research suggests that play which encourages the development of working memory is helpful for preschool children because as it also facilitates the development of emotion knowledge, which is important for social play. The study expands previous perspectives on the mechanisms of the development of preschool children’s emotion knowledge by identifying its mediating influence on the pathway from working memory to social play.

## Figures and Tables

**Figure 1 children-10-00524-f001:**
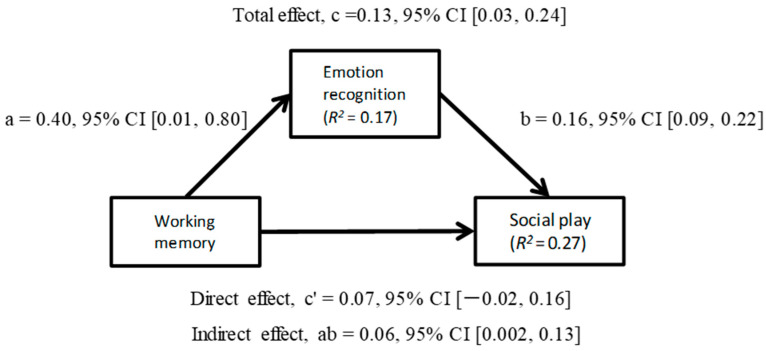
Model of working memory as a predictor of social play, mediated by emotion recognition.

**Table 1 children-10-00524-t001:** Descriptive statistics for the study variables.

Measure	n	*M*	*SD*	Range
Age	47	61.89	6.00	51.0–72.0
Verbal ability	47	19.91	9.37	5.0–44.0
Working memory	47	2.53	0.69	1.0–4.0
Emotion recognition	47	2.19	0.88	0.0–4.0
Situation knowledge	47	3.53	0.72	2.0–4.0
Social play	47	0.40	0.25	0.00–0.90

**Table 2 children-10-00524-t002:** Summary of bivariate correlations.

	1	2	3	4	5	6
1. Gender	-					
2. Age	−0.25	-				
3. Verbal ability	0.02	0.51 **	-			
4. Working memory	0.21	0.16	0.42 **	-		
5. Emotion recognition	−0.06	0.25	0.28 *	0.37 *	-	
6. Situation knowledge	−0.04	0.22	−0.03	−0.06	0.25 ^†^	-
7. Social play	−0.08	0.40 **	0.28 *	0.40 **	0.66 **	0.41 **

† *p* < 0.10, * *p* < 0.05, ** *p* < 0.01.

## Data Availability

Due to the restrictions from the research ethics committee, data sharing is not permitted.

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
