# Peer review of "The Impact of Working Memory on the Development of Social Play in Japanese Preschool Children: Emotion Knowledge as a Mediator"

_children, 2023, doi:10.3390/children10030524_

Round 1

Reviewer 1 Report

Dear authors! For sure you address important mechanisms that broaden scientific understanding of mental processes of preschool children. Still there are important issues that need to be addressed at the moment.

1. It would be great of you would write a longer abstract that would give more idea fo what you are going to describe, particularly in the background part.

2. All mental processes of preschoolers, including cognitive, emotional and social aspects gained a lot of attention of researchers, and continue to attract this attention, but your background section is outstandingly short and uses only few modern papers (from the last 5 years), meanwhile a lot of factors might affect cognitive and emotional development of children, including wide use of gadgets, and probably it would be even more true for Japan with its high technologies.

3. As we all know, development in chilhood has very high rates and 17 months of difference between the youngest and the oldest is a huge gap in development, so it seems irrelevant to put all of them in one sample. Might be, that was the reason you dednt find direct effects.

Author Response

Response to Reviewer 1 Comments

Point 1: It would be great of you would write a longer abstract that would give more idea fo what you are going to describe, particularly in the background part.

Response 1: Following your feedback, I added more information the introduction section.

Lines 9–14:

Through enriched play, children learn social-emotional skills necessary for academic achievement and interpersonal relationships with others. Further research is needed on how specific factors associated with social play, such as working memory and emotion knowledge, interact to promote it. Previous studies have examined the association of working memory and emotion knowledge with social play; however, there are no consistent results as to which abilities influence which skills first.

Point 2: All mental processes of preschoolers, including cognitive, emotional and social aspects gained a lot of attention of researchers, and continue to attract this attention, but your background section is outstandingly short and uses only few modern papers (from the last 5 years), meanwhile a lot of factors might affect cognitive and emotional development of children, including wide use of gadgets, and probably it would be even more true for Japan with its high technologies.

Response 2: Following your feedback, I have added more information to the introduction section.

Lines 28–39:

Play is essential for children’s development of social-emotional competence in early childhood. In particular, social play has been linked to the development of executive function [1], emotion knowledge [2-3], theory of mind [4], social competence [5], and psychosocial adjustment [6], including anxiety and problem behaviors [7], reading comprehension, and math performance [8]. Successful peer relationships in early childhood are also associated with academic achievement and school adjustment beyond elementary school [9–13]. Researchers have found that, among other things, the development of executive function and emotion knowledge has an important influence on socioemotional competence in early childhood [14-15]; however, it is unclear whether one competence developmentally precedes the other. Since play is considered the foundation of the early childhood curriculum [16], it is necessary to examine how executive function and emotion knowledge are related to it.

I have also added more information on:

Social play (Lines 42–49):

Children who interact more with their peers have been reported to have higher language and math skills, less aggressive behavior [18], and better relationships with their teachers [8]. Children with higher levels of imaginative play have been shown to have higher executive function, emotion knowledge, language skills, and more conversations with peers [2,14]. In addition, pretend play is associated with inhibitory function [19-20,1] and social skills [5]. Play with dolls, compared to play with a tablet, may provide more opportunities to rehearse theory of mind and empathy skills and contribute to the development of social-emotional skills [21-23].

Executive function information (Lines 56–69):

EF is a self-regulating cognitive process that controls thoughts, behaviors, and emotions to achieve specific goals. EF develops rapidly in early childhood. Its development not only affects early childhood but also continues into adolescence. Furthermore, the EF is strongly associated with the prefrontal cortex (PFC): EF development is maintained by the growth of neural networks that include the PFC [30]. Children with high executive function have been shown to have better emotional control and less aggressive behavior [31]. Executive function and self-regulation skills have also been shown to interact and have indirect effects on academic achievement and social skills [32-33]. Furthermore, a link between executive function and theory of mind has also been noted [34]. Early executive function predicts theory of mind, and as children age, executive function and theory of mind have been shown to have reciprocal effects [35]. In addition, children with low executive function have been noted to have problems with school adjustment [36], including hyperactivity, inattention, and impulsivity, as well as lower academic achievement [37].

Working memory information (Lines 73–78):

Working memory is a basic EF and is defined as the ability of the mind to hold and manipulate information for complex cognitive tasks. [39]. Children rated as having low working memory have been found to be associated with poorer academic performance [40-41] and lower intelligence [42] compared to those of the same age. Working memory also facilitates children’s development of their own understanding of their mental states, including false beliefs [43] and emotional understanding [44].

Emotion knowledge information (Lines 118–128):

While emotion knowledge predicts higher quality peer relationships, higher social competence, higher academic achievement, and fewer behavioral problems, some research suggests that it is EF that predicts the early development of emotion knowledge [60,61]. Some studies have also shown that low levels of preschool EF predict difficulties with behavioral adjustment in school, especially with hyperactivity, even if the child has moderate social-emotional competence [62]. Thus, some studies suggest that EF is a predictor of emotion knowledge, while others suggest that EF and emotion knowledge interact with each other [15]. Farrell and Gilpin’s study showed emotion knowledge and EFs develop interactively from preschool through kindergarten [15] which means that the ability to identify emotions may help one to inhibit one’s emotions when encountering emotionally challenging situations, and the opposite is true as well.

Point 3: As we all know, development in chilhood has very high rates and 17 months of difference between the youngest and the oldest is a huge gap in development, so it seems irrelevant to put all of them in one sample. Might be, that was the reason you dednt find direct effects.

Response 3: As you noted, the 17-month difference between the youngest and oldest is a large developmental difference. As a result, in this analysis, I followed previous studies and included age as a control variable in the model. Therefore, I do not believe that the issue of an age difference affected the results.

Reviewer 2 Report

The article "The Impact of Working Memory on the Development of Social Play in Japanese Preschool Children: Emotion Knowledge as a Mediator" by Shimizu deals with how working memory elicits social play in japanese preschool children. The article is well-written and well-presented. Introduction is clear and extensive, all the constructs are examined widely. Material and methods are adequately described and the scales used fit the aim of the study. Results are clearly presented and support the conclusion. Discussion is complete and well-referenced.

I would only suggest to check English grammar and spell in some parts.

Author Response

Response to Reviewer 2 Comments

The article "The Impact of Working Memory on the Development of Social Play in Japanese Preschool Children: Emotion Knowledge as a Mediator" by Shimizu deals with how working memory elicits social play in japanese preschool children. The article is well-written and well-presented. Introduction is clear and extensive, all the constructs are examined widely. Material and methods are adequately described and the scales used fit the aim of the study. Results are clearly presented and support the conclusion. Discussion is complete and well-referenced.

Point 1: I would only suggest to check English grammar and spell in some parts.

Response 1: Following your feedback, I asked an English proofreading company to check and proofread my English.

Round 2

Reviewer 1 Report

Thank you for your replies! I believe the paper bacame more reasonable